# Dynamics of Sediment Transport in the Teles Pires River Basin in the Cerrado-Amazon, Brazil

Daniela Roberta Borella [1], Adilson Pacheco de Souza [1,2,*], Frederico Terra de Almeida [2], Daniel Carneiro de Abreu [2,3], Aaron Kinyu Hoshide [3,4], Glauber Altrão Carvalho [5], Rafaela Rocha Pereira [2] and Apoliano Francisco da Silva [2]

[1] Postgraduate Program in Environmental Physics, Federal University of Mato Grosso, Cuiabá 78060-900, MT, Brazil
[2] Institute of Agrarian and Environmental Sciences, Federal University of Mato Grosso, Sinop 78557-287, MT, Brazil
[3] AgriSciences Project, Institute of Agrarian and Environmental Sciences, Federal University of Mato Grosso, Avenida Alexandre Ferronato, 1200, Sinop 78555-267, MT, Brazil
[4] College of Natural Sciences, Forestry and Agriculture, The University of Maine, Orono, ME 04469, USA
[5] Faculty of Engineering, Architecture and Urbanism, and Geography, Federal University of Mato Grosso do Sul, Campo Grande 79070-900, MS, Brazil
\* Correspondence: pachecoufmt@gmail.com; Tel.: +55-66-98136-3805

**Abstract:** The Teles Pires River basin is experiencing significant water challenges due to recent urban growth, expansion of irrigated agriculture, and the rise of hydroelectric power plants in Brazil's forest and savanna regions, impacting water availability and sediment production. This study evaluated and estimated the production of suspended sediment ($Q_{ss}$) and total sediment ($Q_{st}$) in rivers and streams of the Teles Pires River basin, using different sampling methods for suspended-solid discharge: vertical sampling (reference), composite sampling (section), sampling along the standard vertical, and sampling along three verticals, collected using the equal-width increment method. The $Q_{ss}$ and $Q_{st}$ values varied from 0.31 to 39.35 metric tons (t) per day ($d^{-1}$) and from 0.32 to 43.70 t $d^{-1}$, respectively. The average percentages of the entrained solid discharge varied from 3 to 5%, between the dry and rainy seasons, and across all hydrological sections. The different sampling methods of $Q_{ss}$ resulted in similar $Q_{st}$ in each of the monitoring sections. The statistical performance of the simple linear regression model was satisfactory with Willmott index of agreement greater than 0.8234 and 0.9455 for estimates of $Q_{ss}$ and $Q_{st}$, respectively. The dynamics of sediment production and transport was influenced by land use and cover, drainage area, and the hydrological seasonality of the region. The different sampling methods of $Q_{ss}$ are compatible with obtaining suspended and total solid discharge; however, the standard vertical sampling is the most simplified and can be applied in a hydrological section with uniform hydraulic conditions.

**Keywords:** Amazon; equal-width increment sampling; hydro-sedimentology; suspended-solid discharge; Teles Pires River



## 1. Introduction

The erosion process can be accelerated by human activity via the removal of plant cover, intensive mechanized tillage of the soil, and the absence of soil conservation practices, which alter the physical and water characteristics of the soil, making it more susceptible to loss by erosion [1,2]. With this approach, the combined effects of human activity and rainfall characteristics (e.g., intensity, blade, quantity, duration, frequency, and pattern of occurrence) influence the infiltration, storage, and drainage capacity of water in the soil [3,4]. This can result in excessive soil loss from water erosion and the production of sediment in water ways. The processes of transport and deposition of these particles are dynamic and depend on various factors, such as the shape, size, and weight of the particles, in addition to the slope, morphology, and surface runoff speed of the water courses [5,6].

The dynamics of sediment production and transport vary with the region's water seasonality. High intensity of precipitation associated with changes in land use cover enhance the capacity for erosion and sediment transport in more susceptible areas [7], with increasing urbanization, there is a reduction in water infiltration into the soil and an increase in net flow [8], affecting the quantity and quality of sediments transported in watercourses [7,8]. Thus, the sediment load produced in watersheds depends significantly on the amount of precipitated water and land use. Generally, the greatest discharge of sediments occurs in the rainy season when the variation in the net flow is very high compared to the dry season [8], but this is not always a direct relationship, since there are other factors, such as physiographic and geomorphological factors that influence the availability and distribution of sediments in watersheds [8,9].

Due to the increase in population and land occupation, agricultural, civil, industrial and extractive activities are currently essential. Nevertheless, soil conservation must be understood as fundamental to avoid erosion and to control the excessive production of sediment resulting from erosion, and its negative effects on the environment. The transport of high concentrations of sediment has various impacts, including the impoverishment of arable soils with the loss of its surface layer [10], pollution of water bodies, reduction and extinction of aquatic species [11,12], siltation and flooding, changes in the volume and flow of water, morphology changes in riverbeds and watershed margins [10], a reduction in the working life of buildings and reservoirs, and the abrasion of hydroelectric turbines and other equipment [13].

Sedimentation is a natural process that occurs over geological time by the combined or isolated action of physical agents (e.g., water, wind, ice, topography, gravity, and plant cover), and chemical and biological agents (e.g., microorganisms, human activity), fragmenting rocks into smaller particles of varying size [14]. In hydro-sedimentology, the initial process of river sedimentation is soil erosion, caused mainly by the kinetic force of raindrops. This both disaggregates and transports the particles of soil to watercourses [10].

Sediment is transported horizontally and vertically throughout the watercourse. Water characteristics [15] impact the type of material and the particle size of the sediment produced in each watershed [5]. The total solid discharge produced and transported by water ways is represented by fine particles in suspension and by larger particles that are entrained along or deposited on the riverbed. Between these, there is a band of bouncing material called the un-sampled zone, which is nevertheless corrected in calculations of the total solid discharge by equations or in field sampling [15].

On-site determination of suspended-solid discharge is essential due to seasonal variations in sediment transport. Among sampling methods, equal-width increment (EWI) is the most used for sampling suspended sediment due to its simplicity and the greater hydro-sedimentological detail of the cross section [15]. Entrained solid discharge can also be determined from samples, but when equipment, human resources, and finances are limited, all sediment discharge can be estimated and simulated by models consisting of values representing the physical and hydraulic characteristics of the channel [5,16] and through the use of geo-technological tools and resources [17,18].

In Brazil, the study and monitoring of hydro-sedimentological processes are concentrated in the main water courses and in large watersheds and are generally carried out and made available by the National Hydro-meteorological Network (Rede Hidrometeorológica Nacional—RHN), under the coordination of the National Agency for Water and Basic Sanitation (Agência Nacional das Águas e Saneamento Básico—ANA), and the Brazilian Geological Survey (Serviço Geológico Brasileiro—CPRM). In the Amazon, such continuous monitoring has low spatial coverage, and is rare in watersheds with small drainage areas. These limitations are due to logistics, distance, and/or limited access, which makes on-site measurements and the installation and maintenance of equipment for routine monitoring difficult [19].

Brazil's Cerrado-Amazon transition zone lies between the Amazon rain forest to the north and the Cerrado savanna to the south. The Teles Pires River basin is located in

this zone, which is one of the principal agricultural frontier regions in Brazil [20,21]. The Teles Pires River has a high potential for generating hydroelectric energy, since there are already five Hydroelectric Plants (HEPs) installed along its main course. Despite being a region with good surface water availability, there are potential conflicts over water use, especially associated with the generation of energy, irrigation, effluent dilution, and other demands of urbanization. Thus, the development and monitoring of management practices and of land use and occupation in the watershed is of high economic and environmental importance [22].

Little is known about the hydro-sedimentological dynamics of the Teles Pires River, and especially its tributaries, since problems such as the loss of soil, inputs, and fertilizers generated by the erosion process, carry sediment to small tributaries, which are transported to the main outlet, causing the rivers and reservoirs to silt up. In addition to other impacts, this ultimately not only reduces the productive capacity of agricultural land, but also the working volume of reservoirs. Therefore, generating information from this agricultural frontier region in Brazil's Cerrado-Amazon transition zone can aid in the proper handling of water resources, conservation, and soil management in this region.

In the search for operational optimization of field sampling and the costs involved in laboratory analyses for sediment characterization, our objective was to evaluate and estimate sediment production in Teles Pires River sub-basins by: (i) evaluating different sampling for suspended solid discharge: vertical sampling (reference), composite sampling (section), sampling along the standard vertical, and sampling along three verticals; (ii) estimating sediment production for simple linear regressions obtained with different sampling methods suspended and total solid discharge; and (iii) describing sediment production and transport dynamics according the drainage area, types of land use, and hydrological seasonality of the region in each sub-basins of the Teles Pires River.

## 2. Materials and Methods

### 2.1. Study Area and Sampling

Our study area corresponds to the sub-basins within the Teles Pires River basin, located between $7°16'47''$ and $14°55'17''$ S and $53°49'46''$ and $58°7'58''$ W. The Teles Pires River starts in the state of Pará in Brazil and then winds through the state of Mato Grosso, Brazil. The drainage area of the watershed is approximately 141,278.0 km$^2$ and the length of the main waterway is approximately 1498 km. The Teles Pires River basin is located in the agribusiness hub of Mato Grosso, where the predominant plant cover ranges from the Cerrado (upper Teles Pires) and Amazon (middle and lower Teles Pires) biomes (Figure 1). According to the Brazilian Institute of Geography and Statistics (Instituto Brasileiro de Geografia e Estatística—IBGE), the most recurrent soils in the region of the upper Teles Pires are Latosols, Cambisols and Neosols, and in the lower Teles Pires, Argisols, Latosols, Neosols, and Plintosols [23,24].

The predominant climate In the upper Teles Pires and in part of the middle Teles Pires is type Aw (tropical hot and humid), with climate seasonality defined by two hydrological seasons, the rainy season (October to April) and the dry season (May to September). The mean annual precipitation was 1970 mm and mean monthly temperature varied between 24.0 and 27.0 °C from 1972 to 2014, according to Souza et al. [25]. During our evaluation period (2018 to 2021), the study region accumulated average annual rainfall of 1915 mm and average monthly air temperature of 26.8 °C (Figure 2), corroborating what was observed [25].

Measurements of flow and the sampling of suspended and entrained sediment were carried out at monthly intervals from 2018 to 2021, with different collection periods for all eight fluviometric stations (Figure 1 and Table 1). We used an aluminum boat and a 15 hp engine to collect sediment samples in cross-sections when they were more than 1.2 m deep and 10.0 m wide. Sediment samples were stored in plastic pots (1.5 L) and gallons (20 L), in an airy and shaded environment, until processing at the Hydraulic and Hydrology Laboratory of the Federal University of Mato Grosso, in Sinop, Mato Grosso state, Brazil.

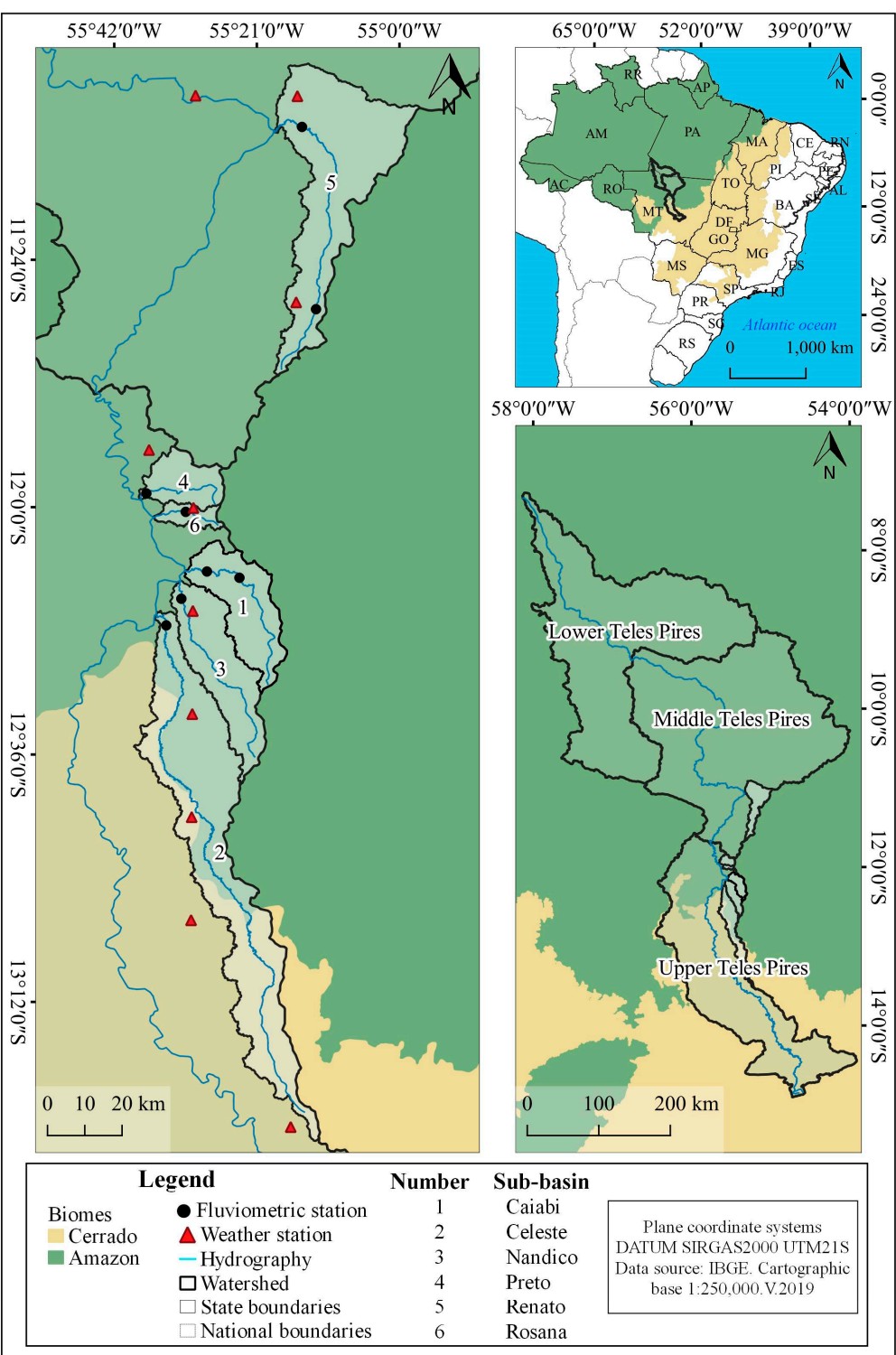

**Figure 1.** Location map of the eight sub-basins of the Teles Pires River basin the Cerrado-Amazon transition zone, Mato Grosso state, Brazil.

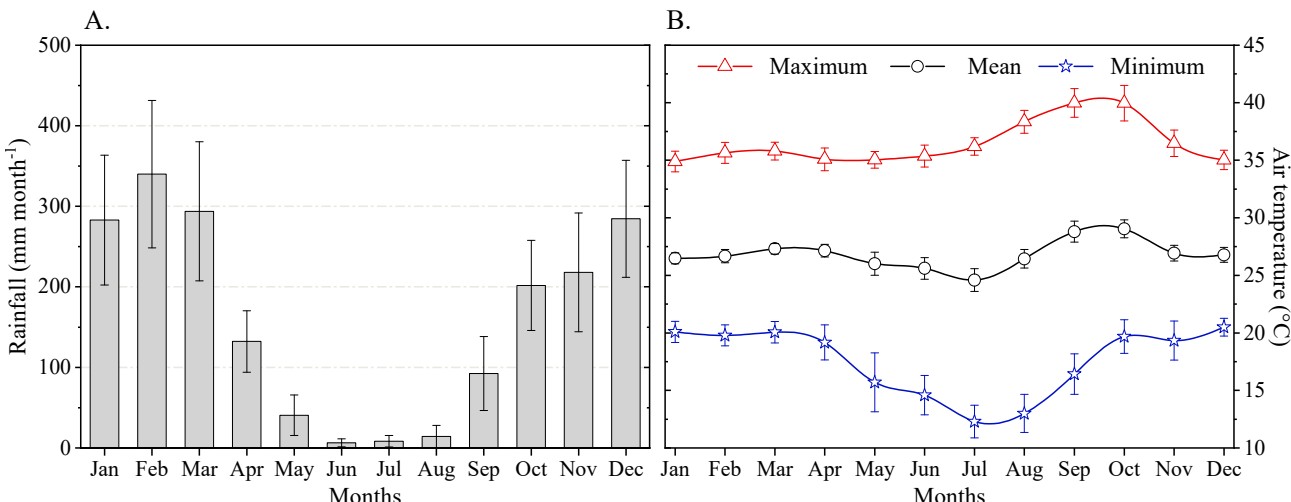

**Figure 2.** Monthly means and standard deviations for rainfall (**A**) and air temperature (**B**) in the study region of the Teles Pires River basin, Mato Grosso, Brazil. Source: Agrometeorological Monitoring System [26].

The general characteristics of the eight sub-basins that we studied are shown in Table 1. In the Caiabi (fluviometric station 1 and 2), Celeste, Nandico, Preto, and Rosana sub-basins, agricultural activities predominate, including the cultivation of soya, maize, cotton, and beans; in the areas of the Preto and Rosana Rivers, there is considerable urban occupation, while in the area of the Renato River (fluviometric station 1 and 2), native vegetation and livestock predominate, with a marked increase in agriculture (Figure 3). Located on the right bank of the Teles Pires River basin, each of the sub-basins has a monitoring section, with the installation of a fluviometric station (Figures 1 and 3). The eight sections for hydro-sedimentological monitoring were defined following hydraulics criteria [27].

**Table 1.** General characteristics of eight sub-basins of the Teles Pires River basin, Mato Grosso, Brazil.

| Sub-Basin | Fluviometric Station | Latitude | Longitude | Altitude (m) | Total Drainage Area (km²) | Hydro-Graphy (km) | Period of Data |
|---|---|---|---|---|---|---|---|
| Caiabi | Caiabi 1 | 12°10′32.64″ S | 55°23′5.22″ W | 372.0 | 492.94 | 62.82 | December 2020 to November 2021 |
| | Caiabi 2 | 12°9′27.23″ S | 55°28′30.39″ W | 345.0 | | | June 2018 to November 2021 |
| Celeste | Celeste | 12°17′39.02″ S | 55°33′56.90″ W | 319.0 | 1800.38 | 220.03 | July 2020 to May 2021 |
| Nandico | Nandico | 12°13′38.11″ S | 55°31′31.45″ W | 334.0 | 589.72 | 63.69 | July 2020 to April 2021 |
| Preto | Preto | 11°58′1.51″ S | 55°37′20.25″ W | 325.0 | 244.44 | 29.59 | May 2020 to May 2021 |
| Renato | Renato 1 | 11°31′18.33″ S | 55°12′12.33″ W | 345.0 | 1336.48 | 94.58 | June 2019 to May 2021 |
| | Renato 2 | 11°4′6.29″ S | 55°14′59.05″ W | 281.0 | | | September 2019 to May 2021 |
| Rosana | Rosana | 12°0′37.36″ S | 55°31′8.13″ W | 332.0 | 84.98 | 20.97 | September 2020 to May 2021 |

The predominant plant cover along the margins of the monitored cross-sections is native vegetation; however, the amount of cover varies according to the width of the watercourse and the type of vegetation, as established in the Brazilian Forest Code (Federal Law no. 12,651 of 25 May 2012). The monitoring sections are close to agricultural areas in the sub-basins of the Celeste, Nandico, and Caiabi Rivers, which are located in the Cerrado biome. In the central region of the sub-basins of the Preto River and Rosana stream, there are significant urban and industrial areas, whereas in the drainage areas of the sub-basin of the Renato River, land occupation by human activity remains low (Figure 3). The conditions of land occupation are important when assessing the seasonality of sediment transport in the region.

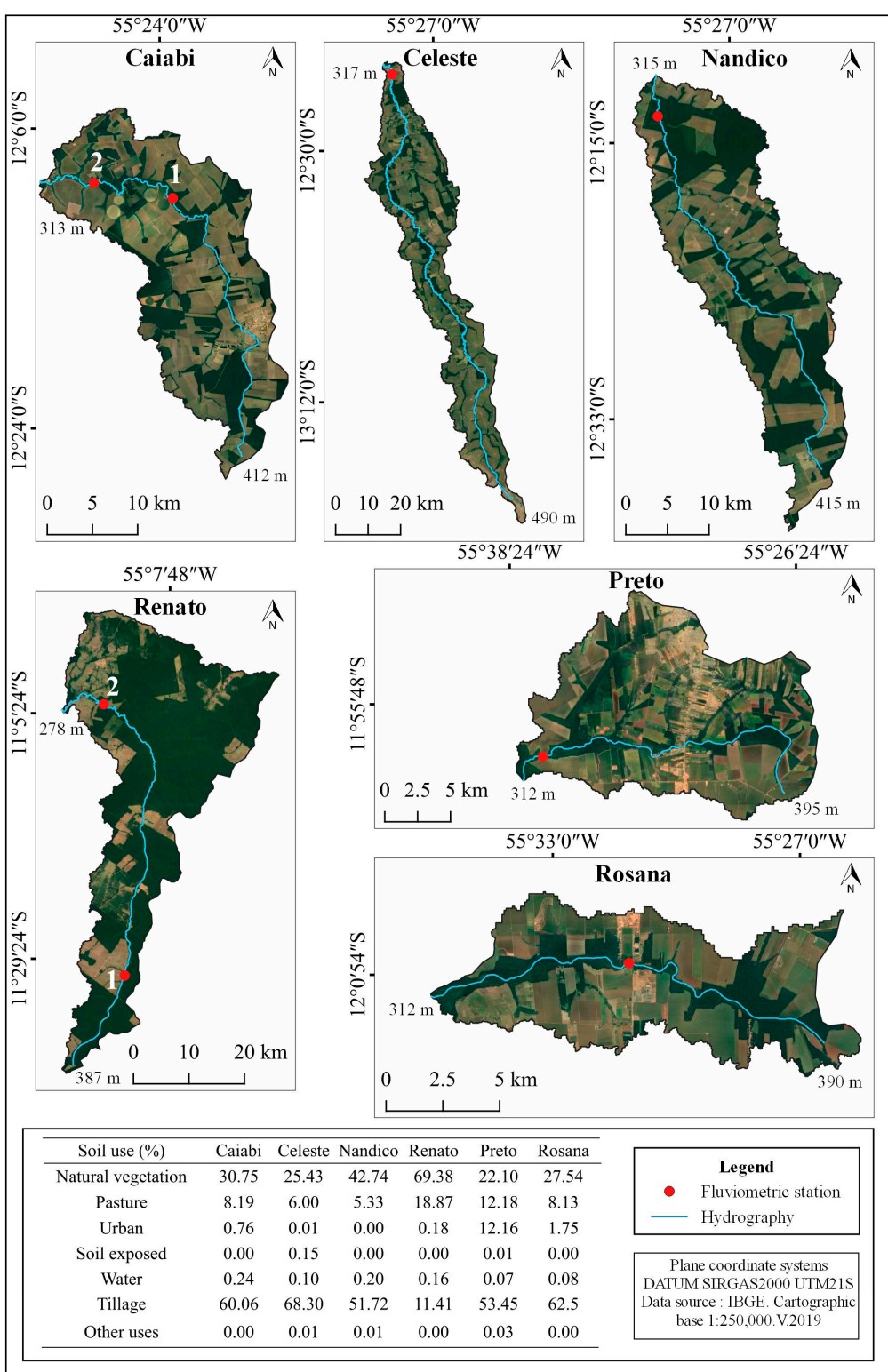

| Soil use (%) | Caiabi | Celeste | Nandico | Renato | Preto | Rosana |
|---|---|---|---|---|---|---|
| Natural vegetation | 30.75 | 25.43 | 42.74 | 69.38 | 22.10 | 27.54 |
| Pasture | 8.19 | 6.00 | 5.33 | 18.87 | 12.18 | 8.13 |
| Urban | 0.76 | 0.01 | 0.00 | 0.18 | 12.16 | 1.75 |
| Soil exposed | 0.00 | 0.15 | 0.00 | 0.00 | 0.01 | 0.00 |
| Water | 0.24 | 0.10 | 0.20 | 0.16 | 0.07 | 0.08 |
| Tillage | 60.06 | 68.30 | 51.72 | 11.41 | 53.45 | 62.5 |
| Other uses | 0.00 | 0.01 | 0.01 | 0.00 | 0.03 | 0.00 |

**Legend**
● Fluviometric station
— Hydrography

Plane coordinate systems
DATUM SIRGAS2000 UTM21S
Data source : IBGE. Cartographic
base 1:250,000.V.2019

**Figure 3.** Location map of the fluviometric stations and principal land uses of the eight sub-basins of the Teles Pires River basin the Cerrado-Amazon transition zone, Mato Grosso, Brazil. Source: land uses [28].

### 2.2. Measurement of Flow

The flow in the eight monitoring sections was measured with a manufacturer JCTM Business and Technology Ltd. (New Bern, NC, USA), a model MLN-7 current meter connected to an electronic revolution counter, and positioned with a graduated metal

wading rod or with a hydrometric winch, depending on the depth of the cross section. The cross section of each sub-basin is kept fixed, adopting the assumptions proposed, which define the number of verticals and points per vertical to be measured based on the width of the cross section and the depth of each vertical [27], respectively, which can vary according to the hydrological season of the year. The total flow was determined by summing the product of the velocity and the wet area of each subsection (Equation (1)) using the half-section method.

$$Q = \sum q_i = \sum (V_i * A_i) \tag{1}$$

where $Q$ is the total flow in cubic meters per second (m$^3$ s$^{-1}$), $q_i$ is the flow rate at each vertical (m$^3$ s$^{-1}$), $V_i$ is the velocity at the subsection in meters per second (m s$^{-1}$), and $A_i$ is the wet area (square meters or m$^2$) of the subsection.

*2.3. Sediment Sampling*

2.3.1. Suspended Sediment

The equal-width increment (EWI) sampling method [15] was adopted when collecting suspended sediment in the eight monitoring sections. In this case, the cross sections were divided into segments or subsections or verticals of equal width, allowing samples of water and suspended sediment to be collected in each subsection or vertical (Figure 4). This method was chosen based on prior knowledge of the velocity distribution and flow of each cross section. In addition, this method is recommended as it simplifies both the methodology in the field and analysis of the results [6,15].

When applying the EWI method, the suspended sediment was measured indirectly, through sampling by integrating along the vertical. This consists of obtaining a sample of water plus suspended sediment along the vertical with uniform transit velocity. The equipment used was the United States Manually operated vertically integrated sampler (US DH-48) sampler with a wading rod, and the United States Vertically integrated sampler (US D-49) coupled to the fluviometric winch. Both samplers have nozzles with a diameter of 3.175, 4.7625 and 6.35 mm (corresponding to diameters of 1/8, 3/16, and 1/4 inches). The un-sampled zone was 0.09 meter (m) and 0.10 m, respectively.

The choice of sampler nozzle depended on the transit velocity, when sampling by integrating along the vertical, was defined by the time taken for the equipment to rise and fall. Collection efficiency tests were then carried out using the above three nozzles, considering the rise and fall along a standard vertical (vertical with the highest product of depth by velocity), using the ratio 1: $V_t/V_m = 0.2$ for the 3.175 mm nozzle and the ratio 2: $V_t/V_m = 0.4$ for the 4.7625 and 6.35 mm nozzles, where $V_t$ is the transit velocity in m s$^{-1}$ and $V_m$ is the mean velocity of the sampled section in m s$^{-1}$. The proper nozzle was the one that collected the maximum mixture of water and suspended sediment, which varied from 3/4 to 4/5 of the volume of the collection bottle, along the standard vertical.

To maintain the same transit velocity (similar that of the water) when sampling by cross-sectional integration, the transit time and sample volume should be different for each vertical (Figure 4). As such, the transit time was calculated for the standard vertical, and later, for the remaining verticals (Equations (2) and (3)):

$$t = \frac{2 * D_v}{V_t} \tag{2}$$

$$t_i = \frac{t * D_i}{D_v} \tag{3}$$

where $V_t$ is the transit velocity (meters/second or m s$^{-1}$); $D_v$ is the depth of the standard vertical (m); and $D_i$ is the depth of the next vertical (m). During sampling, the nozzle of the equipment was kept horizontal, with the opening against the flow of water, avoiding contact with the bed of the river or stream. These precautions avoided errors in the volume of the sample or contamination of the sample from entrained sediment.

Using the same nozzle as used in the efficiency test, the following types of samples of suspended sediment were carried out by integration along the vertical, and shown in Figure 4:

○ Vertical sampling (reference): Consisted in collecting a 1.0 L sample (water and sediment) per vertical cross section, followed by individual storage in sealed plastic bottles identified with the name of the monitoring section, date, sampling method, number of the vertical and transit time.

○ Composite sampling (section): The same procedure as for vertical sampling was repeated. However, the samples from each vertical in the cross section were combined into a single (composite) sample, with an approximate volume of 10.0 L.

○ Sampling along the standard vertical: The collection of a 1.0 L sample along the standard vertical only.

○ Sampling along three verticals: Consisted of collecting a 1.0 L sample along each vertical located at 25% (1/4), 50% (1/2) and 75% (3/4) of the width of the cross section.

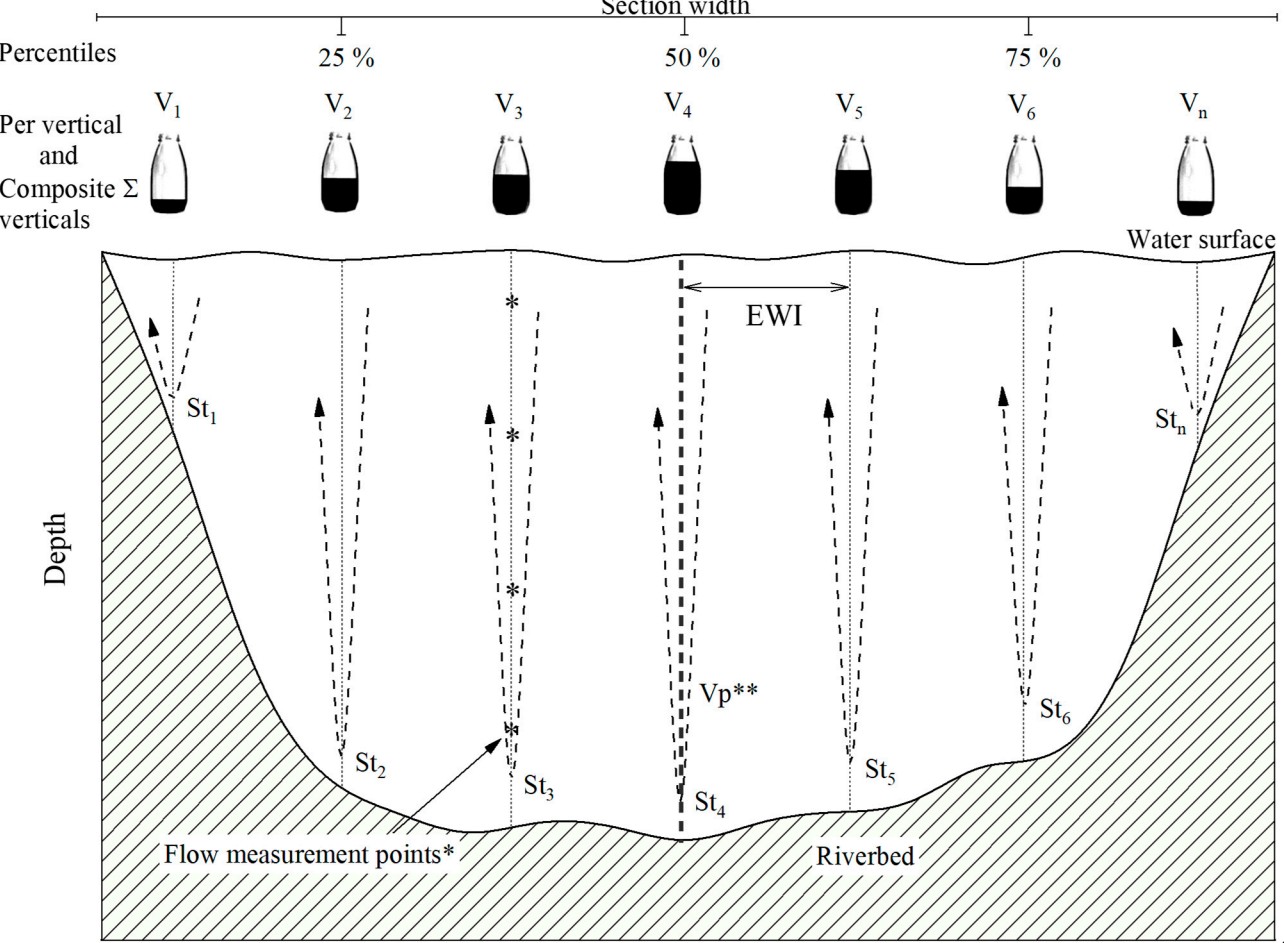

**Figure 4.** Representation of a hypothetical cross section demonstrating the sampling of suspended sediment by integration using the equal-width increment (EWI) method. * Flow measurement points (with fluviometric winch), varying according to the depth of each vertical in the cross section; ** Standard vertical represented by Vertical 4. Representation adapted from Carvalho [15].

A volume of 1.0 L for vertical sampling was defined as sufficient for individual analysis; a volume of 10.0 L for composite sampling was determined based on the suspended-sediment concentration expected for the region, as recommended by the World Meteorological Organization [15].

2.3.2. Entrained Sediment

The entrained sediment was sampled by direct measurement using a United States Manually operated bed discharge samplers (US BLH-84) sampler portable sampler (model Helley Smith) and wading rod. The equipment was placed vertically for 30 min at determined points of the riverbed in the cross section with the opening in the direction of the water flow. The sampler has a sampling efficiency close to 100% for sand and gravel, as it allows entrained sediment to be collected from 0.062 to 16.00 mm using a nylon pouch with a 0.2 mm mesh [15].

The samples of entrained sediment were collected at a location of equal width increment on at least four verticals chosen in alternating positions and distributed along the cross section based on the hydrodynamic characteristics of each monitoring section, always avoiding vertical flows of static or stagnant water. To avoid any loss of sediment, the samples were transferred to plastic bottles, washing the pouch with distilled water. The containers were identified with the name of the monitoring section, date, method of sampling, number of the vertical, and collection time. Samples of water and entrained sediment were stored for later processing in the laboratory.

*2.4. Analysis of the Sediment*

The mean concentration of suspended sediment per sampled vertical and per cross section (i.e., composite sample) was obtained by filtration in a polysulfone vacuum system. This type of system has porous membranes of 0.45 μm and a capacity of 0.5 L per filtration. This method was chosen due to the low sediment concentration found in the rivers and streams under study.

Before filtration, the membranes were dried at 105.0 °C for one hour and transferred to a hermetically sealed silica gel desiccator to be weighed. At the start of each filtration, the samples of suspended sediment were homogenized and filtered. The membranes were dried at 105.0 °C to constant weight, and then kept in the desiccator to avoid the exchange of moisture and to obtain the weight of the set of membranes including the dry sediment. The mean concentration of suspended sediment per vertical and per section was calculated using Equation (4):

$$C_{ss} = \frac{m_2 - m_1}{V} * 1000 \tag{4}$$

where $C_{ss}$ is the concentration of suspended sediment in milligrams per liter (mg L$^{-1}$), $m_1$ is the weight of the clean dry membrane in grams (g), $m_2$ is the weight of the membrane with dry sediment (g), and $V$ is the volume of the sample (L).

The samples of water and entrained sediment from each monitoring section were combined and transferred to metal containers. The organic material was then separated both before and after drying in an oven at 105 °C to constant weight or until all the water had evaporated. The drying and weighing stages of the suspended and entrained sediment were carried out using a forced air circulation oven and a 0.0001 gram (g) precision digital balance, respectively.

*2.5. Calculating the Solid Discharge*

The total solid discharge was calculated as the sum of the suspended-solid discharge sampled along the vertical and the entrained solid discharge (Equation (5)). The entrained solid discharge was calculated using Equation (6), while the suspended-solid discharge was calculated using Equation (7):

$$Q_{st} = Q_{ss} + Q_{se} \tag{5}$$

$$Q_{se} = \left[ \frac{1440 * m * W}{E_{sa} * n * l * t} \right] \tag{6}$$

$$Q_{ss} = 0.0864 * Q * C_{ss} \tag{7}$$

where $Q_{st}$ is the total solid discharge in metric tons per day (t d$^{-1}$), $Q_{ss}$ is the suspended-solid discharge (t d$^{-1}$), and $Q_{se}$ is the entrained solid discharge (t d$^{-1}$) in Equation (5). For Equation (6), $m$ is the total weight of the sample (metric tons or t), $W$ is the width of the cross section (m), $E_{sa}$ is the efficiency of the sampler for 30% of the pouch = 1.0, $n$ is the number of sampled verticals, $l$ is the width of the sampler opening = 0.075 m, and $t$ is the sampling time (minutes). For Equation (7), $Q$ is the total flow (m$^3$ s$^{-1}$) and $C_{ss}$ is the concentration of suspended sediment (mg L$^{-1}$).

Four different types of samples of suspended sediment were evaluated. First, vertical sampling ($Q_{ss}$) was obtained by summing the product of each flow and concentration of suspended sediment along each vertical in the cross section. Second, composite sampling ($Q_{ssc}$) was achieved by summing the product of the flow and total suspended sediment for the entire cross section. Third, sampling along the standard vertical ($Q_{ssvs}$) was obtained from the product of the flow and concentration of suspended sediment along the standard vertical only. The fourth type of sampling was along three verticals ($Q_{ssp}$) from the mean value of the product of each flow and concentration of suspended sediment along each vertical located at 25% (1/4), 50% (1/2) and 75% (3/4) of the width of the cross section.

### 2.6. Estimation Models for Suspended and Total Solid Discharge

The data periods for each monitoring section of the sub-basins were defined by analyzing the consistency and integrity of the measurements of flow, suspended-solid discharge and entrained solid discharge, to determine the total solid discharge. The databases were divided into 70% and 30% of the total data to calibrate the coefficients and evaluate the statistical performance of the estimation models, respectively. Data separation occurred in such a way that all the sub-basins and hydrological seasons of the year (rainy and dry) were represented in each data group (calibration and validation). From this initial separation, a new subdivision was made based on the hydrological seasonality of the region (rainy and dry seasons). For the annual groupings (total data) and hydrological seasons (dry and rainy) simple linear regressions ($y = a + bx$) were calibrated to estimate the mean values of the suspended-solid discharge from the $Q_{ssc}$, $Q_{ssvs}$ and $Q_{ss}$% samples based on the $Q_{ss}$. Estimates for $Q_{st}$ were then obtained considering the different sampling methods for suspended-solid discharge, mentioned above.

To evaluate the performance of the fitted models, the following statistical indicators were used: mean bias error ($MBE$) specified in Equation (8), root mean square error ($RMSE$) defined in Equation (9), and the Willmott index of agreement ($dw$) presented as Equation (10) [29]:

$$MBE = \frac{\sum_{i=1}^{n} Pi - Oi}{n} \tag{8}$$

$$RMSE = \left[ \frac{\sum_{i=1}^{n} (P_i - O_i)^2}{n} \right]^{0.5} \tag{9}$$

$$dw = 1 - \frac{\sum_{i=1}^{n} (P_i - O_i)^2}{\sum_{i=1}^{n} \left( \left| P'_i - \overline{O} \right| + \left| O'_i - \overline{O} \right| \right)^2} \tag{10}$$

where $Pi$ are estimated values, $Oi$ are measured values, and $n$ is the number of observations. In addition, $\left| P'_i - \overline{O} \right|$ is the absolute value of the difference between the estimated value and average of observed values, while $\left| O'_i - \overline{O} \right|$ is absolute value of the difference between the observed value and mean of observed values.

## 3. Results

The profiles of each cross section of the eight sub-basins of the Teles Pires River basin under study showed different geometric and hydrological characteristics between the dry and rainy seasons (Figure 5). The width and average depth of each cross section varied between the maximum and minimum peaks of flow, with the greatest depths seen near the center of each section (except for the Nandico River). The flow showed maximum

peaks between February and April, and minimum peaks between September and October. However, variations in suspended-sediment concentration along the profile of each cross section showed similar behavior between the dry and rainy seasons, except for Rosana and Caiabi (fluviometric station 2) (Figure 5).

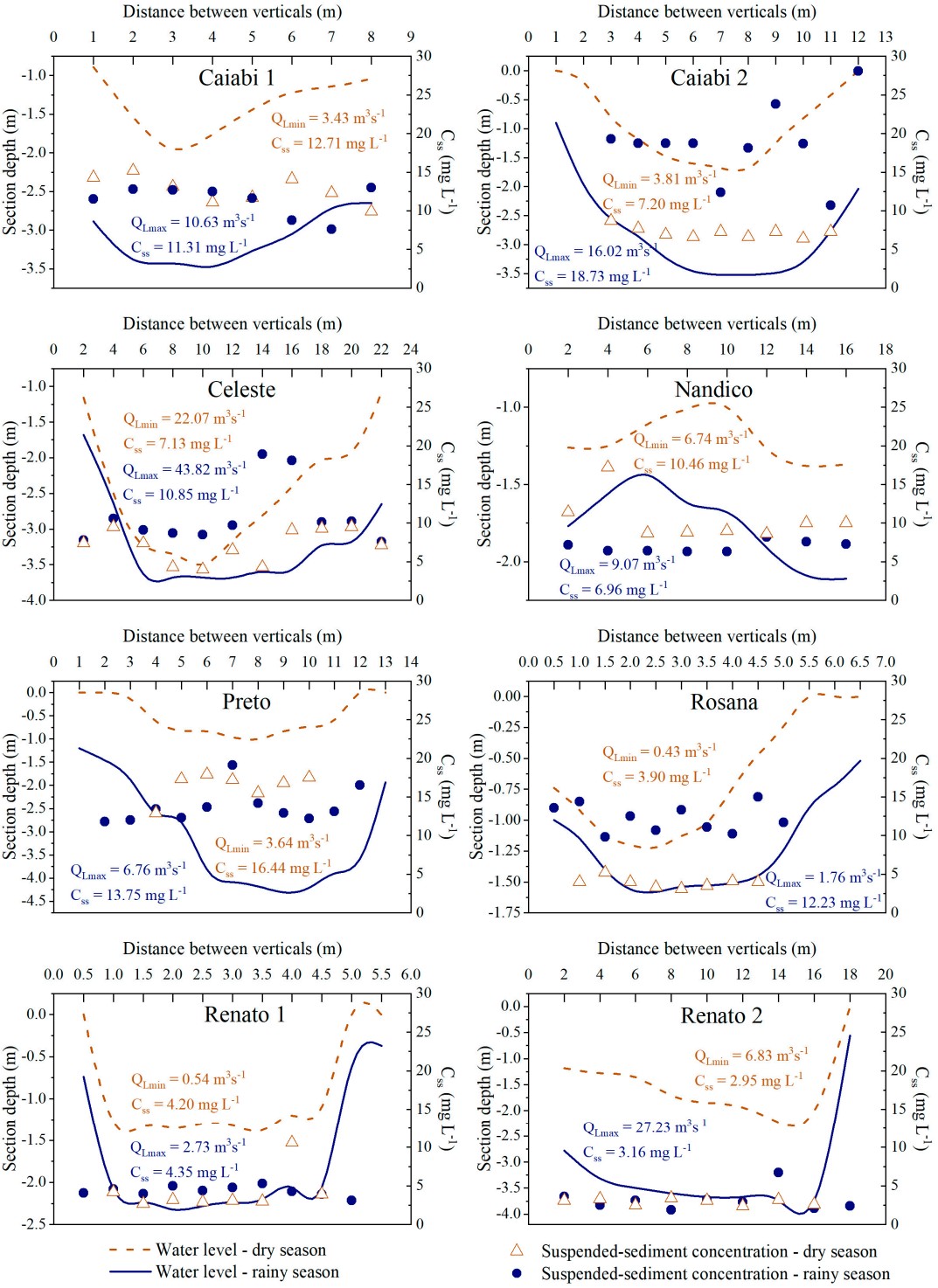

**Figure 5.** Cross-sectional profiles of water level and suspended-sediment concentration ($C_{ss}$) during the dry and rainy season, of monitored cross-sections in eight sub-basins of the Teles Pires River basin, Mato Grosso, Brazil. Note: $QL_{max}$ and $QL_{min}$ are total values for the maximum and minimum flow of each cross section.

Regardless of the monitored cross-sections, the dynamics of flow and solid discharge were similar, with the peaks in suspended, entrained, and total solid discharge following the peaks in flow for the period under evaluation (Table 2 and Figure 6). During the rainy season, there was an increase in surface runoff and solid discharge as the drainage area increases (Table 2 and Figure 6). In addition, the specific total solid discharge is high for the monitoring sections of the Preto River and Rosana Stream compared to the other monitoring sections (Table 2).

We tried to classify the sediment production of each hydrographic sub-basin according to the classification described by Fonseca [7] which varies from very low (<0.0027 metric ton (t) per day ($d^{-1}$) per $km^2$) to extremely high (>0.0548 t $d^{-1}$ per $km^2$). Thus, sediment production in the Preto River is high, while in the Nandico and Renato Rivers (fluviometric stations 1 and 2) it is low, regardless of the water season of the year. The Rosana Stream and the Celeste and Caiabi Rivers (fluviometric station 1 and 2) showed low to moderate sediment production between the dry and rainy seasons in the region, respectively. The low production of sediments in the Renato and Nandico Rivers was due to the conservation of natural vegetation, and, in the other rivers, it was observed that as the soil was occupied by tillage, pasture and urbanization, there was an increase in the production and transport of sediments.

**Table 2.** Mean values flow and solid discharge in different hydrological seasons of the year, in eight sub-basins of the Teles Pires River basin, Mato Grosso, Brazil.

| Hydrological Seasons | Fluviometric Station | Flow ± SD ($m^3 s^{-1}$) | $Q_{ss}$ (t $d^{-1}$) | $Q_{se}$ (t $d^{-1}$) | $Q_{se}$ ± SD (%) | $Q_{st}$ (t $d^{-1}$) | $D_a$ ($km^2$) | $Q_{stesp}$ (t $d^{-1}$ $km^{-2}$) |
|---|---|---|---|---|---|---|---|---|
| Annual | Caiabi 1 | 5.83 ± 2.45 | 4.33 | 0.0948 | 2.31 ± 1.50 | 4.43 | 339.58 | 0.0130 |
| | Caiabi 2 | 9.38 ± 3.48 | 6.18 | 0.2852 | 4.93 ± 4.91 | 6.46 | 454.27 | 0.0142 |
| | Celeste | 28.92 ± 7.89 | 24.84 | 1.8822 | 5.35 ± 3.64 | 26.72 | 1788.24 | 0.0149 |
| | Nandico | 8.02 ± 2.96 | 5.90 | 0.0391 | 0.67 ± 0.35 | 5.94 | 570.22 | 0.0104 |
| | Preto | 5.14 ± 0.98 | 8.58 | 0.1978 | 2.31 ± 2.10 | 8.78 | 242.46 | 0.0362 |
| | Renato 1 | 1.45 ± 0.68 | 0.53 | 0.0040 | 0.81 ± 0.65 | 0.54 | 130.16 | 0.0041 |
| | Renato 2 | 16.24 ± 6.93 | 4.28 | 0.4885 | 9.65 ± 8.94 | 4.77 | 1181.16 | 0.0040 |
| | Rosana | 0.96 ± 0.44 | 0.78 | 0.0207 | 3.16 ± 2.72 | 0.80 | 46.89 | 0.0171 |
| Dry | Caiabi 1 | 3.90 ± 0.47 | 2.80 | 0.0650 | 2.22 ± 1.98 | 2.87 | 339.58 | 0.0084 |
| | Caiabi 2 | 6.20 ± 1.21 | 4.55 | 0.1875 | 3.88 ± 4.39 | 4.74 | 454.27 | 0.0104 |
| | Celeste | 25.20 ± 3.97 | 20.00 | 1.0681 | 3.79 ± 2.41 | 21.07 | 1788.24 | 0.0118 |
| | Nandico | 6.70 ± 0.66 | 5.97 | 0.0405 | 0.69 ± 0.41 | 6.01 | 570.22 | 0.0105 |
| | Preto | 4.70 ± 0.69 | 9.05 | 0.2406 | 2.80 ± 2.16 | 9.30 | 242.46 | 0.0383 |
| | Renato 1 | 0.90 ± 0.20 | 0.31 | 0.0036 | 1.21 ± 0.73 | 0.32 | 130.16 | 0.0024 |
| | Renato 2 | 10.50 ± 2.16 | 3.05 | 0.1527 | 5.05 ± 5.15 | 3.20 | 1181.16 | 0.0027 |
| | Rosana | 0.70 ± 0.14 | 0.59 | 0.0154 | 3.71 ± 3.45 | 0.61 | 46.89 | 0.0130 |
| Rainy | Caiabi 1 | 7.70 ± 2.08 | 5.87 | 0.1245 | 2.40 ± 1.05 | 5.99 | 339.58 | 0.0176 |
| | Caiabi 2 | 11.70 ± 2.57 | 7.36 | 0.3564 | 5.70 ± 5.22 | 7.72 | 454.27 | 0.0170 |
| | Celeste | 40.00 ± 5.37 | 39.35 | 4.3244 | 10.05 ± 2.23 | 43.68 | 1788.24 | 0.0244 |
| | Nandico | 12.00 ± 4.14 | 5.71 | 0.0350 | 0.61 ± 0.06 | 5.75 | 570.22 | 0.0100 |
| | Preto | 6.30 ± 0.38 | 7.31 | 0.0838 | 1.01 ± 1.55 | 7.39 | 242.46 | 0.0305 |
| | Renato 1 | 1.80 ± 0.61 | 0.67 | 0.0043 | 0.56 ± 0.47 | 0.67 | 130.16 | 0.0052 |
| | Renato 2 | 22.90 ± 3.37 | 5.71 | 0.8802 | 15.01 ± 9.79 | 6.59 | 1181.16 | 0.0056 |
| | Rosana | 1.40 ± 0.35 | 1.02 | 0.0274 | 2.48 ± 1.65 | 1.04 | 46.89 | 0.0223 |

Abbreviations: SD = standard deviation, $C_{ss}$ = suspended-sediment concentration, $Q_{ss}$ = suspended-solid discharge, $Q_{se}$ = entrained solid discharge, $Q_{st}$ = total solid discharge, $D_a$ = drainage area from the source to the monitoring section (same across dry and rainy seasons), $Q_{spts}$ = specific total solid discharge, $m^3 s^{-1}$ = cubic meters per second, t $d^{-1}$ = metric tons per day, % = percentage, $km^2$ = square kilometers, and t $d^{-1}$ $km^{-2}$ = metric tons per day per square kilometer.

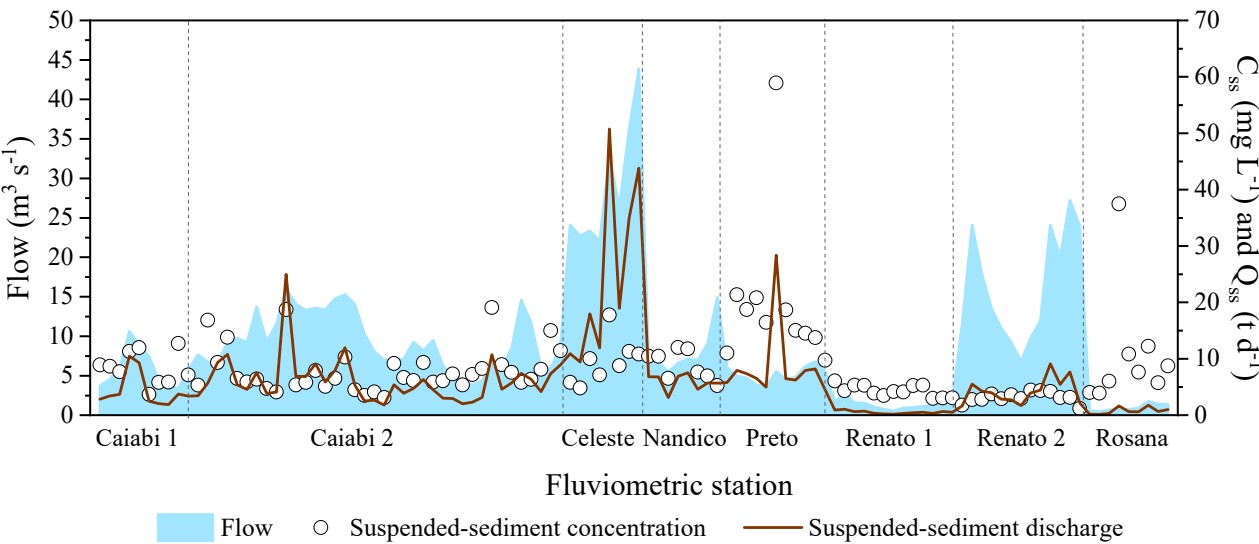

**Figure 6.** Flow, suspended-sediment concentration ($C_{ss}$), and suspended-solid discharge ($Q_{ss}$) measured monthly from 2018 to 2021 in the monitored cross-sections of the eight sub-basins of the Teles Pires River basin, Mato Grosso, Brazil.

Different contributions of entrained solid discharge can be seen in the total solid discharge between the transverse sections of the rivers under evaluation, the annual average varying from 0.67% in the Nandico to 9.65% in the Renato (fluviometric station 2). These variations are more apparent between hydrological seasons. The contribution of entrained sediment to the total discharge increased with the increase in drainage area, as seen for the monitored cross sections of the Caiabi and Renato Rivers, and during the rainy season, for the sections of the Caiabi, Celeste, and Renato (fluviometric station 2) Rivers (Table 2).

The fitted linear regression equations for suspended-solid discharge using different sampling methods and hydrological seasonality showed increasing behavior, with coefficients of determination ($R^2$) greater than 0.93 (Figure 7). In this case, the linear coefficients of the regressions for estimating the suspended-solid discharge were disregarded, since these coefficients would generate residuals for null values of the reference $Q_{ss}$. The values for mean bias error (*MBE*) were lower in the estimates of $Q_{ss}$ that were based on the standard vertical. However, the spread or root mean square error (*RMSE*) was lower for estimates related to each percentile of the cross-sectional profile. Linear regressions generated underestimations of up to 1.88 metric tons per day during the rainy season, in addition to favoring greater spread of the estimates. The Willmott adjustment indices were greater than 0.82, irrespective of the hydrological season (data cluster) or monitoring section (Figure 7).

Regarding the sampling method for suspended sediment, the estimated values closest to the observed values for the reference $Q_{ss}$ were obtained with $Q_{ssp}$ in each of the data clusters except for the Celeste River (Figure 8B–D). In Figure 8, the values of the standard deviations (red lines) indicate the variability of the observed mean values for flow and suspended-solid discharge for different sampling methods and clusters. It was decided to show the observed values (columns) and estimated values (black dots) per monitoring section due to the variability in solid discharge measured in the different cross-sections.

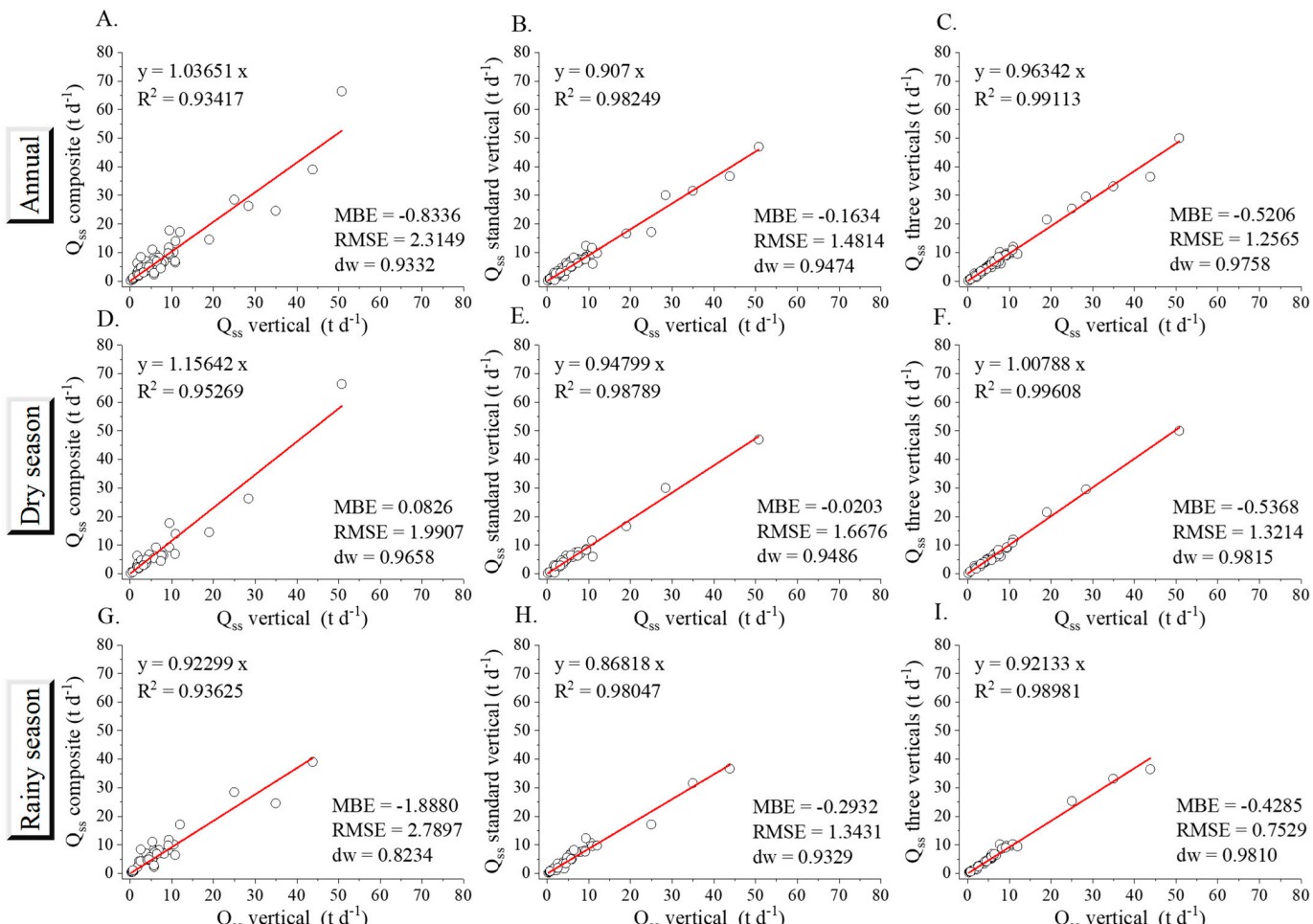

**Figure 7.** Linear estimation equations and their statistical performance for suspended-solid discharge using different sampling methods in the annual grouping (**A–C**), in the dry season (**D–F**), and in the rainy season (**G–I**) in the eight sub-basins of the Teles Pires River basin, Mato Grosso, Brazil. The circles in the figure represents the values of suspended-solid discharge using different sampling methods. Abbreviations: *MBE* = mean bias error, *RMSE* = root mean square error, and *dw* = Willmott index of agreement.

The equations for estimating total solid discharge from the different sampling methods for suspended-solid discharge show improvements for all the statistical indicators of performance (Figure 9). However, total solid discharge is better estimated when the data is grouped by hydrological season. The best fitted equation for estimating $Q_{st}$ during the dry season was by sampling $Q_{ssp}$ with a minimum overestimation of 0.12 metric ton (t) per day ($d^{-1}$). During the rainy season, sampling $Q_{ssc}$, underestimated the total solid discharge by 0.16 t $d^{-1}$ (Figure 9). These errors are negligible when diluted by the drainage areas up to the monitored cross-sections of each tributary of the Teles Pires River under evaluation (Table 2).

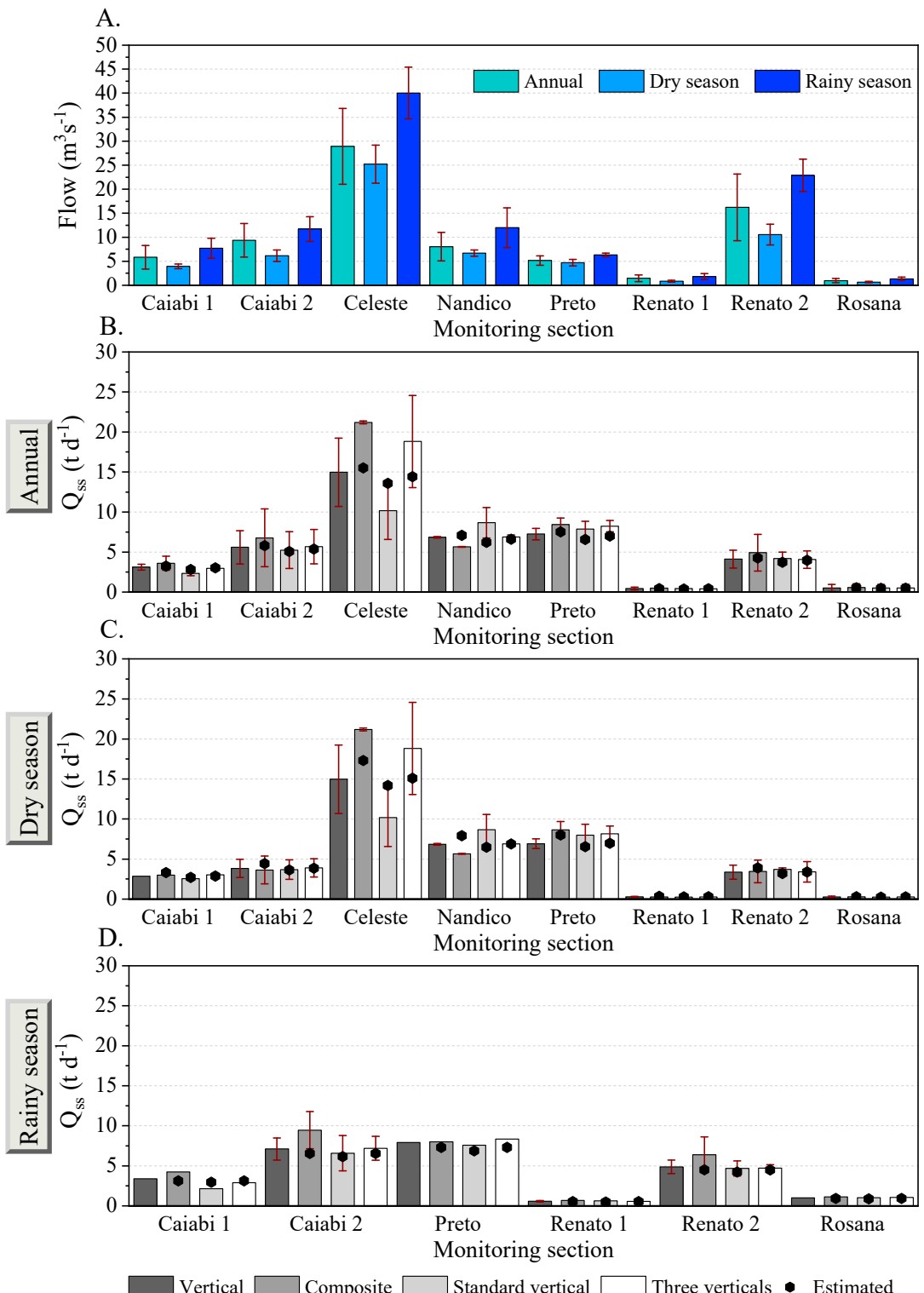

**Figure 8.** Flow (**A**) and suspended-solid discharge, both observed and estimated using different sampling methods, in the annual clusters (**B**), during the dry season (**C**) and during the rainy season (**D**), in the eight sub-basins of the Teles Pires River basin, Mato Grosso, Brazil. Note: The data observed in Figure 8 refer to the 30% separation of the database for statistical validation of the simple linear regression model; the measured values and those estimated by different sampling methods during the rainy season for the monitoring sections of the Celeste and Nandico Rivers are not shown due to the unavailability of measured data.

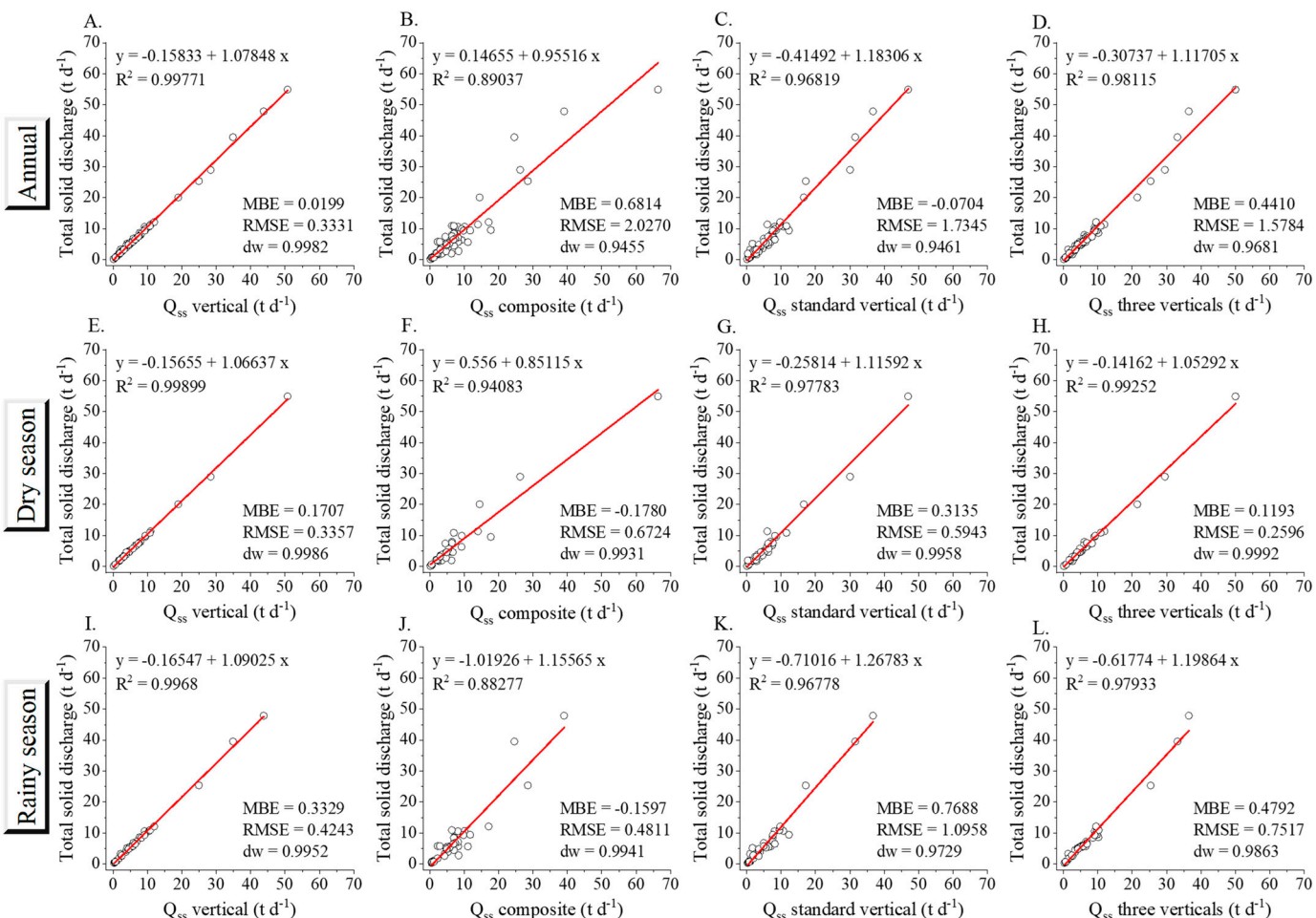

**Figure 9.** Linear estimation equations and their statistical performance for total solid discharge in relation to suspended-solid discharge for different sampling methods in the annual grouping (**A–D**), in the dry season (**E–H**) and in the rainy season (**I–L**) in the eight sub-basins of the Teles Pires River basin, Mato Grosso, Brazil. The circles in the figure represents the values of total solid discharge from the different sampling methods for suspended-solid discharge. Abbreviations: *MBE* = mean bias error, *RMSE* = root mean square error, and *dw* = Willmott index of agreement.

## 4. Discussion

The rivers under study are natural and perennial channels with a flow of continuous surface runoff throughout the year. However, due to the regional hydrological seasons, the volume of drained water and the dynamics of sediment transport vary (Figures 5–7). During the period under evaluation, there was a delay between the periods of maximum (December to May) and minimum (June to November) flow, showing greater discrepancies as the drainage area increased. The rainy season, which starts during the last ten days of September, is responsible for the underground and surface recharge of water in these watersheds. However, the speed of this recharge is dependent on the surface water demand (evapotranspiration), the size of the drainage area, type of soil, the ground cover, management practices, terrain slope, and amount of rainfall [30,31].

The transverse profiles of the monitoring sections showed distinct and stable morphology of the riverbed and margins (Figure 5). The shape of the cross section is responsible for its hydraulic geometry, and influences, above all, the directions taken by the surface runoff. In addition, the geomorphological formation can affect the stability of the bed and margins of a channel [32].

It was apparent in this study that the dynamics of sediment transport were significantly influenced by drainage areas (Table 2). Watersheds with a larger drainage area can transport

more sediment than those with a smaller drainage area [18], however this is not the rule. The Celeste and Renato (fluviometric station 2) rivers have similar drainage areas, but very different values for suspended, entrained, and total solid discharge (Table 2). This can be explained by the predominance of native vegetation in the sub-basin of the Renato River, which affords greater soil conservation, whereas the soils of the sub-basin of the Celeste River are mainly occupied by agricultural crops (Figure 2).

Furthermore, between the Renato (fluviometric station 1) and Rosana sub-basins, which have similar drainage areas up to the sampling points, it was noted that sediment discharge was higher in the Rosana stream. However, in this case, the soils of the sub-basin are more exposed, with greater concentration of agricultural and urban areas, contributing to a greater discharge of sediment from rural roads and/or unpaved streets (Figure 3 and Table 2). In the Preto River and the Rosana stream, which have small drainage areas, the sediment concentrations and total solid discharge per square kilometer ($km^2$) were equal to or greater than those seen in the Celeste River, irrespective of the hydrological season (Table 2 and Figure 6).

The Preto River showed an increase in the concentration and discharge of solids during the dry season (Table 2). This can be explained by the low capacity for sediment dilution of a smaller volume of water during the dry season, for the same conditions of effluent discharge, whereas in the Renato River, the lower concentration and uniform distribution of suspended solids seen along the water course are due to greater soil conservation in the sub-basin, which is occupied by native vegetation (Figures 3 and 6). The Preto River showed important variations in specific total solid discharge, of between 0.0305 and 0.0383 metric ton (t) per day ($d^{-1}$) per $km^2$ of sub-basin area between the wet and dry seasons. It is worth noting that this is a river with considerable urban areas, and therefore receives effluents from industrial processes and sewage networks that help to increase the discharge of solids into the river.

In addition to sediment production in the watersheds, it should be noted that such cases may also be influenced by the volumes of water drained by the channels that are responsible for diluting the suspended solids [18], and also for re-suspending particles during the rainy season [33]. The Celeste River has a greater capacity for diluting suspended solids due to its higher flow. As a result, in the Preto River and Rosana stream, dilution may be compromised by the lower flow.

Agricultural areas with no conservationist management are more susceptible to soil and water loss, and consequently tend to favor an increase in the production of fluvial sediment, especially in the case of a change in soil structure and disaggregation or compaction of the surface layers [1,2]. In the case of the sub-basins of the Preto River and Rosana stream, there was a significant contribution from urban and industrial activities to increase suspended sediment concentrations, with the future possibility of changes in the environmental and sanitary conditions of the water bodies, resulting from water pollution. Other researchers [34,35] recently reported problems related to anthropogenic pollution of urban rivers under different types of land use.

In the eight sub-basins under evaluation, the concentration and discharge of suspended sediment varied by hydrological season, land use, and drainage area (up to the sampling point of each monitoring section) (Figures 3 and 6). Similar behavior was observed by [36], who found a $Q_{ss}$ from vertical sampling of between 1.55 and 22.82 t $d^{-1}$ and a $Q_{st}$ of between 2.0 and 23.0 t $d^{-1}$ in the Caiabi (fluviometric station 2) from 2018 to 2020. Other research [33] reported a large variability in sediment discharge, including an increase during the rainy season, between the four monitoring sections installed in the main watercourse of the watershed for the Jordão River in Minas Gerais. In this study, the suspended solid discharge ranges from 17.31 to 64.64 t $d^{-1}$ and the total solid discharge from 30.57 to 113.83 t $d^{-1}$ for a flow of between 2.4 and 6.7 cubic meters ($m^3$) second ($s$)$^{-1}$, respectively. It can be seen that in the basin of the Jordão River, with its predominance of agricultural activity, sediment production is greater than seen in the watersheds evaluated in our present study (Table 2 and Figure 6).

In the Almas River basin in Goiás, the average sediment production was 0.0088 metric ton (t) per day ($d^{-1}$) per $km^2$ ranging from 0.0027 to 0.0548 t $d^{-1}$ per $km^2$, the greatest variations and sediment production were observed in agricultural areas with agriculture and pasture, exposed soil and steeper areas [7]. Another study reported that in the Guariroba River basin in Mato Grosso do Sul, sediment production increased in the larger drainage area and in steeper areas of the basin; thus, the sediment production in this tropical basin with predominance of pasture and planted forest varied from 0.0331 to 0.0482 t $d^{-1}$ per $km^2$ [37].

While in the Amazon region, the average production of sediments in the Negro river basin was 0.0222 t $d^{-1}$ per $km^2$ with higher volumes in the region's rainy season [18], our study registered values of sediment production close to the cited studies, with average annual production between 0.0040 and 0.0362 t $d^{-1}$ per $km^2$ (Table 2). Considering that they are hydrographic basins with different drainage areas, soil, vegetation, and physiographic characteristics, it is noticed that the land use and cover, as well as the water regime significantly influence the production of sediments.

The increase in sediment discharge during the rainy season may be related to the existence of extensive areas of poorly managed crops and pasture. Studies evaluating water and soil losses in integrated production systems under natural rainfall in this region have shown that land use and management influence soil loss from water erosion. There are also significant reductions in soil and water losses when the forestry component is integrated with crops and/or pasture [38,39], or when no-till and/or minimum tillage cropping systems are adopted [40].

Sediment production in the context of a watershed depends on the interaction between the intensity and duration of the rainfall and the density of the plant cover [3,4], the type of soil, and management practices [1,2,38,39]. In turn, sediment transport is the result of the slope of the channel, the increase in flow speed, and the increase in flow, especially during the rainy season [6,33]. Studies on the dynamics of suspended sediment transport carried out in the Negro River in the Amazon [18] and the Jordão River in the Mineiro Triangle and Upper Paranaíba [33] affirm that sediment concentration and discharge are controlled by the water regime and land use. Surface runoff influences the dynamics of sediment transport [16] and transport capacity may increase with an increase in runoff flow [6].

It is worth pointing out that the present study proposed to evaluate whether the sampling method for suspended sediment can be simplified and optimize with statistical confidence, which would allow the activities of field collection and the costs associated with laboratory analyses. In fact, the different sampling methods such as composite, along the standard vertical, and along the three verticals at 25, 50, and 75% of the width of the cross section, gave results that were very close to those of vertical sampling. Furthermore, in the regional context of the Teles Pires River basin, the simple linear regression mathematical model that was used estimated the suspended and total solid discharge with accuracy.

The sampling method for suspended sediment had little influence on the mean deviations or spread of the estimates for suspended and total solid discharge, allowing a good fit and good statistical performance when applying simple linear regression, irrespective of the hydrological season in the region. It is important to note that the effect of the variability in mean deviation on the suspended solid discharges obtained with the different sampling methods (collections), monitoring sections, and hydrological seasonality was to be expected. This is because there are differences in the morphological and water characteristics of the cross sections, drainage areas, and types of land use in each of the sub-basins under study.

There is no universal equation for sediment transport available or recommended for wide use, since each empirical model has its own range of application [16]. This is the case with the fitted regressions in the present study, which considered the suspended sediment obtained with different sampling methods and their influence on total sediment production. Applying these equations to other watersheds is limited, as they depend on the physical and water characteristics and on the sediment production and transport dynamics of each

cross section. It is necessary first to understand the hydro-sedimentological behavior of the section before choosing the sampling method and equation.

Studies should be carried out that group a large number of watersheds by drainage area, land use, or sediment production and transport with similar dynamics. This would allow calibration and validation of new models for estimating suspended and total solid discharge. These studies would be fundamental in planning land and water use, as well as in decision-making and implementing effective practices of soil management, with the aim of reducing losses from erosion [1], pollution, and the siltation of water bodies [12].

## 5. Conclusions

Sediment production and transport dynamics in rivers and streams of the Teles Pires River basin depend on the hydrological seasonality of the region, the drainage area, and the types of land use in each sub-basin. Simple linear regressions are suitable for estimating the relationship between suspended solid discharge and total solid discharge obtained with different sampling methods. The mean percentages for entrained solid discharge from the rivers and streams of the Teles Pires River basin vary from 3 to 5% between the hydrological seasons in the region. The different sampling methods for suspended sediment are compatible with obtaining suspended and total solid discharge and should be considered in future occasions for other watersheds. However, sampling along the standard vertical is considered the most simplified, as it is the hydro-sedimentological reference vertical of a cross section.

The rivers and streams of the Teles Pires River basin have low temporal and spatial availability of hydro-sedimentological information, similar to what occurs in other Amazonian hydrographic basins. In this region, routine monitoring programs for creating and maintaining databases must be maintained and prioritized. This will allow future applications for defining reference hydrological variables (ecological flows, residence time, extremes of flows, among others) and hydro-sedimentological processes (production, transport, and accumulation) in the region.

**Author Contributions:** Data collection, writing, methodology, formal analysis, figures—D.R.B. and A.P.d.S.; data collection, review, editing, supervision—A.P.d.S. and F.T.d.A.; review, editing, supervision, and financial support—A.K.H. and D.C.d.A.; supervision, data collection—G.A.C., R.R.P. and A.F.d.S. All authors have read and agreed to the published version of the manuscript.

**Funding:** This study was financed by the Coordenação de Aperfeiçoamento de Pessoal de Nível Superior—Brasil (CAPES) and the Agência Nacional de Águas e Saneamento Básico (ANA), Finance Code—001 and Process 88887.144957/2017-00. The authors wish to thank the Conselho Nacional de Desenvolvimento Científico e Tecnológico (CNPq) for their support with scientific initiation grants and a productivity grant (Process 308784/2019-7).

**Institutional Review Board Statement:** Not applicable.

**Informed Consent Statement:** Not applicable.

**Data Availability Statement:** Study data can be obtained by request to the corresponding author or the first author, via e-mail. It is not available on the website as the research project is still under development.

**Acknowledgments:** The authors thank the 3ª Comarca de Justiça Cível de Sinop for their financial support of the Projeto CBH Monitora. The Comitê dos Afluentes da Margem Direita do Alto Teles Pires for the institutional and technical-scientific commitment. The authors also thank all the students and professors of the Tecnologia em Recursos Hídricos no Centro-Oeste research group (dgp.cnpq.br/dgp/espelhogrupo/2399343537529589).

**Conflicts of Interest:** The authors declare no conflict of interest. Supporting entities had no role in the design of the study; in the collection, analyses, or interpretation of data; in the writing of the manuscript, or in the decision to publish the results.

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
