# Peer review of "Dynamics of Sediment Transport in the Teles Pires River Basin in the Cerrado-Amazon, Brazil"

_sustainability, doi:10.3390/su142316050_

Round 1

Reviewer 1 Report

This study examines and estimates the production of sediment in rivers and streams of the Teles Pires River basin using different sampling methods. The paper is well organized and quite aptly written. For further improvement, authors should consider the following comments in the revised version.

1. The authors carried out several tasks but these are not reflected in the objectives mentioned in the introduction. I would suggest listing the objectives in bullet points at the end of the introduction.

2. The authors should compare the average production of sediments to the same with other notable rivers in that region or beyond. That would reflect how serious the subject matter is. The authors should include that in the discussion section.

3.  The Teles Pires River basin should be included in the title of the paper

4.  Figure 7 and 9: Please add parameters in both Y and X axis

Author Response

This study examines and estimates the production of sediment in rivers and streams of the Teles Pires River basin using different sampling methods. The paper is well organized and quite aptly written. For further improvement, authors should consider the following comments in the revised version.

This study examines and estimates the production of sediment in rivers and streams of the Teles Pires River basin using different sampling methods. The paper is well organized and quite aptly written. For further improvement, authors should consider the following comments in the revised version.

  1. The authors carried out several tasks but these are not reflected in the objectives mentioned in the introduction. I would suggest listing the objectives in bullet points at the end of the introduction.

We have listed the major objectives of our research at the end of the Introduction as follows:

As included between the lines 121 at 127.

  1. The authors should compare the average production of sediments to the same with other notable rivers in that region or beyond. That would reflect how serious the subject matter is. The authors should include that in the discussion section.

We have contrasted the sediment levels we found in the Teles Pires River basin to other river basins in the region and beyond. Specifically, we have contrasted with the following research:     

  • Line 541 at 548 - In the Almas River basin in Goiás, the average sediment production was 0.0088 ton (t) day (d)-1 per km2 ranging from 0.0027 to 0.0548 ton (t) day (d)-1 per km2, the greatest variations and sediment production were observed in agricultural areas with agriculture and pasture, exposed soil and steeper areas (7). Another study reported that in the Guariroba River basin in Mato Grosso do Sul, sediment production increased in the larger drainage area and in steeper areas of the basin, thus, the sediment production in this tropical basin with predominance of pasture and planted forest varied from 0.0331 to 0.0482 ton (t) day (d)-1 per km2 (37).
  • Line 549 at 555 - While in the Amazon region, the average production of sediments in the Negro river basin was 0.0222 ton (t) day (d)-1 per km2 with higher volumes in the region's rainy season (18). Our study registered values of sediment production close to the cited studies, with average annual production between 0.0040 and 0.0362 ton (t) day (d)-1 per km2 (Table 2). Considering that they are hydrographic basins with different drainage areas, soil, vegetation and physiographic characteristics, it is noticed that the land use and cover, as well as the water regime significantly influence the production of sediments.
  1.  The Teles Pires River basin should be included in the title of the paper

We have changed the title as suggested to “Dynamics of sediment transport in the Teles Pires River Basin in the Cerrado-Amazon, Brazil”

  1.  Figure 7 and 9: Please add parameters in both Y and X axis

Figures 7 and 9 corrected.

Reviewer 2 Report

The article concerns the dynamics of sediment transport in the rivers. It may be interesting for readers of Sustainability. In general, this manuscript is well organized and written, with comprehensive literature review, detailing the framework approach of the study, clearly stated methodology and nicely presented findings. The discussed research problem is particularly important in the aspect of progressive climate change. The following requests/suggestions should be taken into account to improve the quality of the manuscript.

-  Please explain how the hydrological regime of rivers, including changes in water flow rate, affect the dynamics of sediment transport. Such information would certainly expand knowledge in the field of processes affecting water quality.

-  Some of the methods you use are known, please highlight your achievements in this work.

-  Is it possible to determine the rate of accumulation of sedimentary matter based on the presented method?

Author Response

The article concerns the dynamics of sediment transport in the rivers. It may be interesting for readers of Sustainability. In general, this manuscript is well organized and written, with comprehensive literature review, detailing the framework approach of the study, clearly stated methodology and nicely presented findings. The discussed research problem is particularly important in the aspect of progressive climate change. The following requests/suggestions should be taken into account to improve the quality of the manuscript.

-  Please explain how the hydrological regime of rivers, including changes in water flow rate, affect the dynamics of sediment transport. Such information would certainly expand knowledge in the field of processes affecting water quality.

We have added this background information to the Introduction section and specifically we have cited the following:

  • Lines 49 at 59 - The dynamics of sediment production and transport vary with the region's water seasonality. High intensity of precipitation associated with changes in land use cover enhance the capacity for erosion and sediment transport in more susceptible areas [7], with increasing urbanization there is a reduction in water infiltration into the soil and an increase in net flow [8], affecting the quantity and quality of sediments transported in watercourses [7,8]. Thus, the sediment load produced in watersheds depends significantly on the amount of precipitated water and land use. Generally, the greatest discharge of sediments occurs in the rainy season when the variation in the net flow is very high compared to the dry season [8], but this is not always a direct relationship, since there are other factors, such as physiographic and geomorphological factors that influence the availability and distribution of sediments in watersheds [8, 9].

-  Some of the methods you use are known, please highlight your achievements in this work.

We have specified the following unique contributions to the methodologies that we used:

“The authors aimed, with the different methods of sampling suspended sediments, to simplify and optimize the operational work in the field and the costs associated with laboratory analyses. This information was inserted between the lines 575 at 578.”

-  Is it possible to determine the rate of accumulation of sedimentary matter based on the presented method?

The authors believe that in the future it will be possible to define accumulation rates in the studied rivers and in the watershed of the Teles Pires river, however, not only with the results obtained in this article.

The article evaluated sediment sampling methods for the characterization of instantaneous transport in a hydrological section. Sediment accumulation will depend on the total annual production of the watershed. Accumulation analyzes depend on the adjustments of the solid flow key-curves for the hydrological sections and also on the routine (continuous) monitoring of variables associated with sediment production (eg. turbidity). Thus, our article sought to evaluate methods that will allow the future obtainment of solid key-curves (which need a greater number of years).

Any extrapolation of the data obtained in this work needs to be evaluated with caution, as the dates of measurements/collections did not necessarily represent the maximum and minimum limits of suspended sediment concentrations and entrained (bottom). Therefore, the present article is an initial and important step for future analyzes of “accumulation/deposition” of sediments in the evaluated rivers.

Reviewer 3 Report

The present manuscript entitled “Dynamics of sediment transport in the rivers of the Cerrado-Amazon, Brazil” is a quality research conducted by the authors. I appreciated the hard work and research concerns of the authors. However, despite the importance of the idea, the manuscript language in most section is confused. Moreover, I highlighted some changes in different sections of the manuscript, which may further aid in the quality of the manuscript. In addition the literature in the manuscript is not sufficient, more literature needed to be added and cites. Therefore, I will recommend major revision with thorough text filtration for improvement.

Title: Title is well addressed and reflects the manuscript

Abstract

The abstract lacks the quantitative evaluation, results and conclusion of the research.

- Line 17. Replace “given” by “due to”

-Line 19. Delete “both” please

-Line 21. Different sampling methods, very general please specify

-Line 25-30. Confusing results, try to write in simple way if possible

-Conclusion is missing

-Look into keywords also please

Introduction

The section needs restructuring clearly mentioning the research importance, gaps and objective of the current research

-I will advised rearrangement of first three paragraphs for coherence in the sections i.e. the 3rd paragraph can be made 1st and the 1st paragraph can be made 3rd.

-The research gap and objective can better be evaluated if possible

-Thorough editing is needed along with grammar check

Material and method

The methodology is comprehensive but confusing by not justifying the analysis used in plan sequence

Well and comprehensive written but need topographic correction.

-Line 160. JCTM model MLN-7, explain when 1st time used as abbreviation

-Line 177 Please enclosed “EWI” in bracket and delete “or”

-Line 187. US DH-48, explain when 1st time used as abbreviation

-Line 191. Delete “which,” please

-Line 317. Please use the abbreviations properly

Results

The results looks good, but its like presenting same data and results again and again. It make confusion please be specific and clear in your results for the readers

-Line 331-340. Unnecessary paragraph. If included better to fit it in methodology study area

-Table 2 and Figure 6. Did the data different or same data presented two times

- Add legends for the units in table also i.e. t d-1 and others

-Line 379. Mean square error (MAE)???

-Figure 7. Add legends for the abbreviation dw, MAE,RMMS..

-Figure 8 and Table 2 data, is not it the same data presented again

Discussion

The section is well written and appreciated

-Line 445. Again I see plant cover and its importance but the data or its measurement is lacking in results and methodology.

Conclusion

The conclusion can be improved and also add limitation of the study if needed.

Author Response

The present manuscript entitled “Dynamics of sediment transport in the rivers of the Cerrado-Amazon, Brazil” is a quality research conducted by the authors. I appreciated the hard work and research concerns of the authors. However, despite the importance of the idea, the manuscript language in most section is confused. Moreover, I highlighted some changes in different sections of the manuscript, which may further aid in the quality of the manuscript. In addition the literature in the manuscript is not sufficient, more literature needed to be added and cites. Therefore, I will recommend major revision with thorough text filtration for improvement.

Title: Title is well addressed and reflects the manuscript

Abstract

The abstract lacks the quantitative evaluation, results and conclusion of the research.

We have modified the abstract as requested

- Line 17. Replace “given” by “due to”

We have corrected this typographical error

-Line 19. Delete “both” please

We have corrected this typographical error

-Line 21. Different sampling methods, very general please specify

We specified this more specifically as “vertical sampling (reference), composite sampling (section), sampling along the standard vertical and sampling along three verticals” Line 22-23

-Line 25-30. Confusing results, try to write in simple way if possible

We have simplified this part of the results presented in the Abstract and re-written as “The different sampling methods of Qss resulted in similar Qst in each of the monitoring sections, the statistical performance of the simple linear regression model was satisfactory with Willmott index of agreement greater than 0.8234 and 0.9455 for estimates of Qss and Qst, respectively.” Line 26-29

-Conclusion is missing

We have added this as a final sentence to the Abstract as “The dynamics of sediment production and transport was influenced by land use and cover, drainage area, and the hydrological seasonality of the region. The different sampling methods of Qss are compatible with obtaining suspended and total solid discharge, however the standard vertical sampling is the most simplified and can be applied in a hydrological section with uniform hydraulic conditions.” Line 29-33

-Look into keywords also please

We have clarified and edited the keywords used.

Introduction

The section needs restructuring clearly mentioning the research importance, gaps and objective of the current research

-I will advised rearrangement of first three paragraphs for coherence in the sections i.e. the 3rd paragraph can be made 1st and the 1st paragraph can be made 3rd.

            We have switched the 3rd paragraph in the Introduction with the 1st paragraph as suggested to improve coherence

-The research gap and objective can better be evaluated if possible

            Reviewer 1 has also brought up this concern, and we have edited this.

-Thorough editing is needed along with grammar check

            We have done another complete round of edits including for grammar

Material and method

The methodology is comprehensive but confusing by not justifying the analysis used in plan sequence

Well and comprehensive written but need topographic correction.

-Line 160. JCTM model MLN-7, explain when 1st time used as abbreviation

We specified this as “manufacturer  JCTM Business and Technology Ltda, model MLN-7” Line 194-195

-Line 177 Please enclosed “EWI” in bracket and delete “or”

            We have made this edit

-Line 187. US DH-48, explain when 1st time used as abbreviation

            We specified this as “United States Manually operated vertically integrated sampler (US DH-48)” Line 221-222

            We specified this as “United States Vertically integrated sampler (US D-49)” Line 222-223

We specified this as “United States Manually operated bed discharge samplers (US BLH-84)” Line 271-272

-Line 191. Delete “which,” please

            We have made this edit

-Line 317. Please use the abbreviations properly

We have made these edits including clarifying the abbreviation for mean square error as “MSE” and not “MAE”

Results

The results looks good, but its like presenting same data and results again and again. It make confusion please be specific and clear in your results for the readers

-Line 331-340. Unnecessary paragraph. If included better to fit it in methodology study area

            We have moved this paragraph to the Methods section

-Table 2 and Figure 6. Did the data different or same data presented two times

Table 2 presents the average values of flow and solid discharge in different hydrological stations of the evaluated period.

In the Table 2 was deleted the Css and was maintain only in Figure 6, and to facilitate understanding the Table 3 was deleted and its information included in Table 2.

We have included information between lines 385 and 395 and moved the paragraph about the results from Table 3 between lines 396 and 402 before Table 2.

Figure 6 shows monthly instantaneous values of liquid and solid flow in each monitored cross-section.

- Add legends for the units in table also i.e. t d-1 and others

We have made this edit

-Line 379. Mean square error (MAE)???

The authors used the MBE (mean bias error), as it allows analyzing the relative differences between observed and estimated values, allowing association with under or overestimates. When squaring or adopting absolute values, there is a perception of the general error of the model and not of the relative mean differences.

-Figure 7. Add legends for the abbreviation dw, MAE,RMMS..

Correction by MBE

-Figure 8 and Table 2 data, is not it the same data presented again

“The data shown in Figure 8 are not the same as those shown in Table 2. The data in Figure 8 refer to the 30% separation of the database for statistical validation of the simple linear regression model. This information was included in the footer of the figure.”

“Replaces the figure to standardize writing of legend.

Discussion

The section is well written and appreciated

-Line 445. Again I see plant cover and its importance but the data or its measurement is lacking in results and methodology.

We have deleted discussion related to natural vegetation reducing erosion and thus suspended sediment since land use was not directly measured with our research.

Conclusion

The conclusion can be improved and also add limitation of the study if needed.

We have specified the limitations of our research study as follows and added this to the end of the Conclusion:

Lines 620 at 626 - “The rivers and streams of the Teles Pires River basin have low temporal and spatial availability of hydro-sedimentological information, similar to what occurs in other Amazonian hydrographic basins. In this region, routine monitoring programs for creating and maintaining databases must be maintained and prioritized. This will allow future applications for defining reference hydrological variables (ecological flows, residence time, extremes of flows, among others) and hydro-sedimentological processes (production, transport and accumulation) in the region.

Round 2

Reviewer 3 Report

The Authors have substantially revised the manuscript and can be accepted in the present form.